# Association Between the Renin–Angiotensin System and Ibrutinib-Related Cardiovascular Adverse Events: A Translational Cohort Study

**DOI:** 10.3390/biomedicines13092184

**Published:** 2025-09-06

**Authors:** Jonaz Font, Amir Hodzic, Angélique Da-Silva, Baptiste Delapierre, Ghandi Damaj, Anne Neusy, Anne-Flore Plane, Damien Legallois, Paul Milliez, Charles Dolladille, Mégane Vernon, Sarah Burton, Nicolas Vigneron, Christophe Denoyelle, Joachim Alexandre

**Affiliations:** 1INSERM U1086 ANTICIPE, Université de Caen Normandie, Normandie Université, Biology-Research Building, Avenue de la Côte de Nacre, F-14000 Caen, France; jnzfnt@gmail.com (J.F.); dasilva-a@chu-caen.fr (A.D.-S.); anne.neusy@orange.fr (A.N.); damien.legallois@unicaen.fr (D.L.); charles.dolladille@unicaen.fr (C.D.); 2Cardiac Electrophysiology Unit, Cardiology Department, Caen-Normandy University Hospital, Avenue de la Côte de Nacre, F-14000 Caen, France; paul-ursmar.milliez@unicaen.fr; 3UMR 1075 COMETE, Université de Caen Normandie, Normandie Université, 2 Rue Des Rochambelles, F-14032 Caen, France; amir.hodzic@unicaen.fr; 4Departments of Cardiology and Clinical Physiology, Caen-Normandy University Hospital, Avenue de la Côte de Nacre, F-14000 Caen, France; 5PICARO Cardio-Oncology Program, Departments of Pharmacology and Medical Oncology, Caen-Normandy University Hospital, Avenue de la Côte de Nacre, F-14000 Caen, France; 6Hematology Institute, Caen-Normandy University Hospital, Avenue de la Côte de Nacre, F-14000 Caen, France; delapierre-ba@chu-caen.fr (B.D.); damaj-gl@chu-caen.fr (G.D.); 7PICARO Cardio-Oncology Program, Department of Cardiology, Caen-Normandy University Hospital, Avenue de la Côte de Nacre, F-14000 Caen, France; plane-af@chu-caen.fr; 8GIP Cyceron, INSERM U1237 PhIND, Université de Caen Normandie, Normandie Université, Boulevard Henri Becquerel, F-14000 Caen, France; 9PICARO Cardio-Oncology Program, Department of Pharmacology, Caen-Normandy University Hospital, Biology-Research Building, Avenue de la Côte de Nacre, F-14000 Caen, France; 10Interdisciplinary Research Unit for Cancers Prevention and Treatment, BioTICLA Laboratory (Precision Medicine in Ovarian Carcinoma), Université de Caen Normandie, Inserm, ANTICIPE UMR (1086), Federative Structure 4207 Normandie Oncologie, F-14000 Caen, France; m.vernon-contentin@baclesse.unicancer.fr (M.V.); sarah.burton@unicaen.fr (S.B.); n.vigneron@baclesse.unicancer.fr (N.V.); christophe.denoyelle@unicaen.fr (C.D.); 11Comprehensive Cancer Center F. Baclesse, Unicancer, 3 Avenue Général Harris, F-14000 Caen, France; 12Calvados General Tumor Registry, Comprehensive Cancer Center F. Baclesse, Unicancer, 3 Avenue Général Harris, F-14000 Caen, France

**Keywords:** ibrutinib, atrial fibrillation, hypertension, heart failure, B-cell malignancies, renin–angiotensin system

## Abstract

**Background:** Ibrutinib has been associated with an increased risk of cardiovascular adverse events (CVAEs), including atrial fibrillation (AF), hypertension (HTN), heart failure (HF), and ventricular arrhythmias (VAs). However, baseline predictors of CVAEs remain poorly characterized. In this study, we sought to identify baseline patient characteristics associated with the occurrence of ibrutinib-related CVAEs, with particular emphasis on parameters linked to the renin–angiotensin system. **Methods**: We conducted a prospective, single-center cohort study of consecutive patients treated with ibrutinib for B-cell malignancy, with systematic assessment of a predefined panel of potential predictors of CVAEs at baseline (NCT03678337). These predictors included demographic and clinical variables, 16 circulating biomarkers related to inflammation, fibrosis, and neurohormonal activation, as well as nine echocardiographic parameters. The primary objective was to evaluate the association between baseline patient characteristics and the occurrence of CVAEs from ibrutinib initiation through the end of follow-up. The CVAE endpoint was defined as a composite of atrial fibrillation, new or worsening hypertension, new or worsening heart failure, and ventricular arrhythmias. Statistical analyses were performed using the Wilcoxon–Mann–Whitney test or Fisher’s exact test, with a *p*-value < 0.05 considered statistically significant. **Results:** Among the 25 patients included, 7 experienced a total of 9 CVAEs over a median follow-up of 672 days. Elevated baseline plasma renin levels (>1336.10 pg/mL) were significantly associated with CVAEs occurrence (57% vs. 11%, *p* = 0.032). Higher baseline plasma aldosterone levels (>488.95 pg/mL) were also observed in patients who developed CVAEs, although this association did not reach statistical significance (*p* = 0.058). **Conclusions**: Baseline plasma renin level was univariably associated with CVAEs occurrence, while plasma aldosterone levels were higher among patients with CVAEs but did not reach statistical significance. These findings provide preliminary insights into the mechanisms underlying ibrutinib-related cardiovascular toxicity, suggesting a potential role for the renin–angiotensin–aldosterone system. Confirmation of this hypothesis, however, will require larger, dedicated studies.

## 1. Introduction

Ibrutinib, a Bruton’s tyrosine kinase inhibitor (BTKi), is approved for the treatment of B-cell malignancies [1]. In chronic lymphocytic leukemia (CLL), ibrutinib has demonstrated a significant survival benefit, reducing the risk of progression or death by 84% compared with chlorambucil after a median follow-up of 18.4 months [2].

Nevertheless, it is now well-known that ibrutinib is associated with an increased risk of several cardiovascular adverse events (CVAEs), including atrial fibrillation (AF), new or worsening arterial hypertension (HTN), new or worsening heart failure (HF), and ventricular arrhythmia (VA) [3]. For instance, a safety meta-analysis found an increased risk of HTN with ibrutinib with a relative risk (RR) of 2.82 (95% CI: 1.52–5.23) and an increased risk of AF with an RR of 4.69 (95% CI: 2.17–7.64) [4]. These CVAEs are associated with significative morbimortality. In a retrospective cohort study of 298 CLL patients where 16% developed ibrutinib-related AF, patients with AF had lower progression-free survival (hazard ratio (HR) = 2.0, 95% CI: 1.1–3.8, *p* = 0.02) and shorter overall survival (HR = 3.2, 95% CI: 1.6–6.3, *p* = 0.001) [5].

Although ibrutinib-related CVAEs are now encountered daily in clinical practice, predictive factors are not known; thus, no predictive score exists to predict CVAEs in the ibrutinib-exposed population. ESC cardio-oncology guidelines recommend the evaluation of every patient with a cardio-oncologic examination before BTKi initiation to stratify the baseline risk of developing cardiovascular toxicity [3]; however, to date, the only characteristics that seem associated with ibrutinib-related CVAEs are age, history of HTN, and left atrium dilatation [6].

In the context of ibrutinib-related CVAEs, early identification of patients at risk is essential. The aim of this study was therefore to assess baseline clinical, echocardiographic, and biological characteristics associated with the development of CVAEs in a prospective, single-center cohort of patients treated with ibrutinib for B-cell malignancies.

## 2. Materials and Methods

### 2.1. Study Protocol and Population

This study was nested within the PICARO cohort (NCT03678337), a prospective observational study. We included adult patients with active CLL referred for a baseline cardio-oncology evaluation at Caen–Normandy University Hospital (France) either prior to, or within one month of, ibrutinib initiation between 6 December 2018 and 1 April 2021 [7]. Non-adult and adult protected patients, pregnant or nursing women, and patients under guardianship, curatorship, safeguard of justice, or legal protection were not included. The study protocol was compliant with the STROBE Statement [8].

All patients underwent a baseline cardio-oncology evaluation before or within 1 month after the introduction of ibrutinib according to the ESC cardio-oncology guidelines including a careful clinical history and physical examination, a 12-lead ECG, a blood examination including cardiac biomarkers, and a standard transthoracic echocardiography [3]. Blood peripheral collection tubes were collected at the end of the clinical evaluation, with the patients resting quietly in a semi-recumbent position. Blood samples were collected in EDTA2K tubes, rapidly centrifuged at 2300× *g* for 11 min, and then plasma was aliquoted and stored at −80 °C.

### 2.2. Measurement of Levels of Plasma Protein Biomarkers by ELISA

Commercially available enzyme-linked immunosorbent assays (ELISAs) were utilized to quantify the levels of C reactive protein (CRP) (R&D Systems, Abingdon, UK), Galectin-3 (R&D Systems), Myeloperoxydase (R&D Systems), renin (R&D Systems), aldosterone (ThermoFisher Scientific, Illkirch-Graffenstaden, France), Tumor Necrosis Factor alpha (TNF-α) (R&D Systems), interleukin (IL-6) (R&D Systems), angiotensin converting enzyme 2 (ACE-2) (ThermoFisher Scientific), and troponin (Abcam, Cambridge, UK) according to the manufacturer’s instruction. For routine analyses (serum electrolytes, creatinine…), commercially available kits were used.

### 2.3. Measurement of Expression of Serum microRNAs (miRNAs) by RT-qPCR

We selected six microRNAs (miR-9, miR-99, miR-199, miR-328, miR-22, and miR-150-5p) that have previously been associated with CVAEs or with cardiovascular disease in populations other than patients treated with ibrutinib [9,10,11,12,13,14]. MiRNAs were isolated from samples according to the manufacturers’ recommendations using the NucleoSpin miRNA Plasma kit (Macherey-Nagel, Hoerdt, France). As previously described, known quantities (200 attomoles in 5 µL) of three exogenous miRNAs, cel-miR-39-3p, cel-miR-54-3p, and cel-miR-238-3p, were added to the samples after the denaturing step [15]. The expression of each miRNA was determined by RT-qPCR. miRNAs were first retrotranscribed using an miRNA Reverse Transcription Kit (ThermoFisher Scientific). The ID references for stem-loop primers and Taqman hydrolysis probes (ThermoFisher Scientific) were as follows: cel-miR-39-3p (000200), cel-miR-54-3p (001361), cel miR 238-3p (000248), cel-miR-9-5p (000583), cel-miR-22-5p (002301), cel-miR-99b-5p (00436), cel-miR-150 5p (000473), cel-miR-199a 5p (00498), and cel-miR 328 3p (00543). Fluorescence and threshold baselines were measured using a Roche LightCycler^®^ 480 system with the LightCycler^®^ 480 Software version 1.5.1.62 SP3 (Roche Diagnostics, Meylan, France). Absolute standard curves were generated according to the MIQE guidelines by diluting synthetic cel-miRs and previous cited miRNAs at 2.10^5^, 2.10^4^, 2.10^3^, 2.10^2^, 2.10^1^, and 2 zmol/mL prior to quantitative PCR with reverse transcription (RT-qPCR) steps as previously described [16]. They were used to convert quantitative cycles (Cq) into a log of quantities and then into miRNA concentrations expressed in zeptomoles per microliter of RNA extracts. Serum isolation yields of cel-miR-39, cel-miR-54-3p, and cel-miR-238-3p were calculated for each sample by dividing their recovered quantities by their added quantities. For each sample, serum miRNA concentrations were estimated in zeptomoles per microliter by dividing their recovered quantity by the geometric mean of the three cel-miR yields.

### 2.4. Follow-Up

#### 2.4.1. Blood Pressure Monitoring

Blood pressure was measured every 6 months during the systematic consultation and out-of-the-office measurement was conducted systematically every week by the patient themselves during the follow-up.

#### 2.4.2. ECG Monitoring

A 10-s 12-lead ECG was carried out during every semestrial consultation. Out-of-the-office 5-day ECG monitoring was systematically carried out every 6 months during the follow-up.

#### 2.4.3. Echocardiography

Echocardiography was systematically performed every year during the follow-up. The echocardiographic assessment of left ventricular (LV) and left atrial (LA) functions was performed in accordance with current guidelines, using a commercially available echocardiographic system (Epiq 7 equipped with an X5-1 xMATRIX-array transducer, Philips, Netherlands, Amsterdam) [17]. All data were stored digitally, and offline analysis was conducted using the Philips IntelliSpace workstation. The LV ejection fraction (EF) was calculated using the Biplane Modified Simpson’s method. Diastolic function was evaluated from transmitral E and A velocities, E/A ratio, average of the septal and lateral annular Ea velocity, and E/Ea ratio [17]. Doppler parameters were obtained as the average value of three consecutive cardiac cycles. LA volume was estimated by the Biplane method of disks at end-systole from apical 4-chamber and 2-chamber views. LA dilatation was defined as an LA volume > 34 mL/m^2^ and if not available area > 20 cm^2^ by apical 4-cavity view [18]. Two-dimensional-speckle tracking measurements of LA phasic strains were performed following current practice recommendations [19]. The LA reservoir, conduit, and contractile strains were calculated with the first reference frameset at the onset of the QRS-wave of the surface ECG [19]. Values were obtained from a single apical 4-chamber view. Indexing LA reservoir strain to E/Ea was used as a surrogate for LA compliance, as previously described [20].

### 2.5. Primary Objective and Analysis

The primary objective of this study was to investigate the association between baseline patient characteristics (demographics, clinical, circulating biomarkers expression, echocardiography parameters) and CVAEs occurrence from ibrutinib introduction to 31 May 2023, the end of follow-up. CVAEs outcome was a composite of AF, new or worsening HTN, new or worsening HF, and ventricular arrhythmias [3]. AF was defined as supraventricular tachyarrhythmia with uncoordinated atrial electrical activation lasting at least 30 s or lasting an entire 12-lead ECG [21]. HTN was diagnosed if in-the-office blood pressure was superior or equal to 180 mmHg for systolic blood pressure and superior or equal to 110 mmHg for systolic blood pressure. Otherwise, for in-the-office systolic blood pressure measured between 140 mmHg and 179 mmHg and/or in-the-office diastolic blood pressure measure between 90 and 109 mmHg, diagnosis of HTN was performed with out-of-the-office blood pressure measures with means >135 mmHg for systolic blood pressure and/or >85 mmHg for diastolic blood pressure [22]. In patients with a history of HTN, the need of anti-hypertensive treatment up-titration for hypertension indication counted as an event. HF was defined as a first or recurrent unplanned hospitalization or urgent visit for HF or need to introduce/increase the dose of loop diuretic [23]. VA was defined as any sustained (lasting at least 30 s) ventricular arrhythmia documented on a 12-lead ECG or on a Holter-ECG [24].

### 2.6. Secondary Objectives and Analyses

Each of the components of the primary outcome was described as well as death from any cause and B-cell malignancy progression during follow-up.

Time to CVAE onset and severity by Common Terminology Criteria for Adverse Events (CTCAE) of CVAEs were described.

Description of treatment adjustments, involving ACE inhibitors, ARBs, and anticoagulant therapies, were described.

### 2.7. Statistical Analyses

A descriptive analysis of the cohort study was performed on the available data. Quantitative variables were expressed as median and interquartile range. Qualitative variables were expressed in effectives and percentages. Comparative analyses were performed between patients with and without CVAE occurrence during follow-up. The biological variables were dichotomized using the threshold that maximized Youden’s index, identifying the value with the greatest discriminative ability between groups (Appendix A) [16]. The non-parametric Wilcoxon–Mann–Whitney test was used for quantitative variables, and Fisher exact test was used for qualitative variables. Kaplan–Meier curves and log-rank test were performed for significant qualitative variables. Variance inflation factors (VIFs) were calculated for variables that were significant in the univariate analysis to assess collinearity. Statistical significance was defined as a *p*-value of <0.05. Statistical analyses were performed with RStudio software V.4.2.1 for Windows using the following packages: data.table, dplyr, fst, ggplot2, kableExtra, lubridate, pROC, questionr, rlang, stringr, survival, survminer, tidyverse.

## 3. Results

### 3.1. The Study Population

From 6 December 2018 to 12 March 2021, a total of 25 consecutive patients referred for a baseline cardio-oncological evaluation were included before or within 1 month after the introduction of ibrutinib for B-cell malignancy. The median follow-up was 672 days (588–738) and nine patients stopped ibrutinib for non-cardiovascular causes during this follow-up. Baseline characteristics of the included patients are presented in Table 1. The median age was 72 (63–77) years old, 64% of patients were male, 36% had a prior history of hypertension, and only one patient had a prior history of atrial fibrillation. All included patients had a baseline LVEF > 60% and no patient had a previous history of HF. Ibrutinib was introduced mainly for chronic lymphocytic leukemia (68%). At baseline, 16% of the patients were being treated with beta-blockers, 24% with aspirin, 8% with anticoagulants, 12% with angiotensin-converting enzyme inhibitors, 16% with angiotensin-receptor blockers, 32% with dihydropyridine calcium-channel blockers, and 20% with statins.

### 3.2. Primary Objective

During the follow-up, nine CVAEs occurred in 7 of the 25 patients. Among the 9 CVAEs observed, there were two cases of new-onset HTN and four cases of worsening HTN, along with two cases of new-onset AF and one recurrence of AF. No new or worsening HF or VA was reported.

The characteristics of the patients at baseline according to CVAEs occurrence are presented in Table 1. Patients who experienced CVAEs were significantly more at baseline on dihydropyridine calcium-channel blockers (71.4 vs. 16.7%), had non-significant higher body mass index (24.6 vs. 22.9 kg/m^2^), non-significantly higher HTN history (71.4% vs. 22.2%), and had a non-significantly higher E/Ea ratio (9.9 vs. 8.1; *p* = 0.083). Patients who experienced CVAEs had significant higher plasmatic renin (57% vs. 11% with renin > 1336.1 pg/mL; *p* = 0.032) and lower miR-150-5p (57% vs. 100% with miR-150-5p > 27.38 zmol/µL; *p* = 0.015) levels at baseline (Figure 1). Furthermore, creatinine levels (43% vs. 6% with creatinine > 100 µmol/L), aldosterone levels (71% vs. 22% with aldosterone > 488.95 pg/mL), and troponin levels (43% vs. 6% with troponin > 127.2 pg/mL) were higher in patients experiencing CVAEs without reaching statistical significance (Table 1).

The survival analysis with Kaplan–Meier curves and log-rank test confirms the association between baseline HTN, baseline plasmatic renin level, and baseline plasmatic miR-150-5p level and CVAEs occurrence with respective *p*-values of 0.013, 0.024, and <0.0001 (Figure 2).

All VIF values were below 5, specifically 1.262, 1.262, and 1.000, respectively for baseline HTN, baseline plasmatic renin level, and baseline plasmatic miR-150-5p level.

### 3.3. Secondary Objectives

During the follow-up, five patients experienced B-cell malignancy progression and five patients died from non-cardiovascular causes. Among the 9 CVAEs, only one AF event was classified as severe with a CTCAE = 3 (Figure 3). The median time to onset of CVAEs was 181 (136–492) days.

Among patients who experienced one or more CVAEs, direct oral anticoagulants were initiated in three patients for atrial fibrillation, and an ACE inhibitor was initiated in one patient. In the group without CVAEs, a direct oral anticoagulant was initiated in one patient for an indication other than atrial fibrillation, and the ACE inhibitor dosage was reduced in one patient.

## 4. Discussion

Ibrutinib, a first-in-class irreversible oral inhibitor of BTK, has proven highly effective in B-cell malignancies [2]. These malignancies are usually diagnosed in elderly patients in whom frequent cardiovascular comorbidities coexist at diagnosis that increase, independently of ibrutinib exposure, the risk of CVAEs [25]. Some baseline patient characteristics were previously associated with CVAEs occurrence in patients exposed to ibrutinib such as age > 65, history of HTN or AF, and left atrial dilatation or strain but these comorbidities are well-known risk factors for cardiovascular diseases in the general population [6,26,27]. Therefore, today there is no specific baseline predictive factor associated with ibrutinib-related CVAEs.

In this prospective cohort, although monocentric and including a small number of patients, we systematically assessed a large number of baseline patient characteristics including demographic, clinical, biological, and echocardiographic parameters. Particularly, we designed a large panel of biological variables to study inflammation, fibrosis, and neurohormonal pathways, which represent major signaling pathways in the development of HTN, AF, and HF [28]. Importantly, baseline renin levels were associated with CVAEs occurrence in our cohort and baseline aldosterone levels were higher in patients with CVAEs occurrence, although not statistically significant. It is now well established that the renin–angiotensin–aldosterone system (RAAS) and plasmatic aldosterone levels are associated with HTN, AF, and HF development [29]. Its role in hypertension is notably due to the vasopressive effect of angiotensin-2 and due to the sodium reabsorption effect in the renal tubule [30]. Aldosterone is also known to promote AF independently of HTN. Our group previously demonstrated that patients presenting with primary aldosteronism had a significantly higher rate of cardiovascular events than matched patients affected by essential hypertension [31]. Interestingly, AF was diagnosed in 7.3% of patients with primary aldosteronism and 0.6% of patients with essential hypertension; therefore, primary aldosteronism was associated with a 12-fold higher prevalence of AF compared with essential hypertension. Plasmatic aldosterone levels were previously associated with the development of AF, HTN, and HF in different settings and populations [32,33,34]. Moreover, there is evidence that aldosterone and the activation of its receptor, mineralocorticoid receptor (MR), promote atrial fibrosis and inflammation and modify cardiac action potential by interacting with ion channels and therefore can create a substrate for AF development [35]. A meta-analysis that included 7914 patients showed that mineralocorticoid receptor antagonists (MRAs) were associated with a significantly lower AF risk compared with no MRA treatment (15.0% versus 32.2%; odds ratio, 0.55; 95% CI, 0.44–0.70 [*p* < 0.00001]) [36]. MRAs are effective therapies for managing HTN (e.g., spironolactone) and HF with reduced ejection fraction (e.g., spironolactone and eplerenone), where they reduce overall mortality. More recently, MRAs such as finerenone have also shown beneficial effects on HF outcomes in patients with mildly reduced or preserved ejection fraction [22,23,37,38]. Interestingly, MRA efficacy does not seem confined to patients with HF [38].

There are several reasons to believe that aldosterone and the activation of its receptor, MR, might play an important role in the underlying pathophysiology of ibrutinib-related CVAEs occurrence [39]. The underlying mechanisms of ibrutinib-related CVAEs are not yet fully elucidated but it is believed that the binding of ibrutinib to off-target kinases (e.g., other than BTK) at therapeutic concentrations may potentially contribute to the development of CVAEs [40]. This hypothesis is strongly corroborated by observations in a mouse model where inhibition of the C-Terminal Src Kinase (CSK) by ibrutinib was identified as the causal mechanism of ibrutinib-related AF [41]. CSK has previously been linked to HTN, cardiac fibrosis, and inflammation in populations not exposed to ibrutinib. Furthermore, a cardiovascular magnetic resonance imaging study indicated that ibrutinib exposure induces fibrosis and inflammation of the left atrium, potentially increasing susceptibility to atrial fibrillation [42]. Moreover, inflammation, fibrosis, HTN, and AF induced by aldosterone and its receptor, MR, appeared to be mediated by the inhibition of CSK, resulting in the activation of Src in several experimental models; therefore, the CSK–Src–aldosterone–MR axis emerged as a potential axis involved in cardiovascular diseases (Figure 4). Indeed, in mice, reduced CSK activity has been shown to increase CYP11B2 (aldosterone synthase) expression in the zona glomerulosa of the adrenal gland, leading to elevated aldosterone levels [43]. Moreover, CSK inhibition results in enhanced Src activation, which not only amplifies angiotensin II production via the MAPK/ERK signaling pathway but may also contribute to increased renin release, potentially through the modulation of sympathetic signaling or juxtaglomerular cell function [44]. Taken together, these observations support a potential role for the CSK–Src–aldosterone–mineralocorticoid receptor (MR) axis in the development of ibrutinib-related CVAEs. Although hypothesis-generating and requiring confirmation in future studies, it is conceivable that MRAs could be effective in preventing or treating CVAEs in patients with B-cell malignancies receiving ibrutinib, particularly non-steroidal MRAs, which are emerging as safer and potentially more efficacious than steroidal MRAs [43,45,46,47].

Our analysis highlighted that baseline plasmatic miR-150-5p was significantly lower in patients who experienced ibrutinib-related CVAEs. MiRNAs are small non-coding RNAs involved in genetic expression regulation at a post-transcriptional level [48]. MiR-150-5p is known to regulate cellular proliferation and cardiac fibrosis and was previously associated with AF in the general population [10]. In a prospective study comparing 112 AF patients with 99 controls without AF, miR-150-5p was associated with AF with a significantly greater expression in the control group after adjustment (*p* = 0.04) [10]. Kawaguchi et al. demonstrated in both mice and humans that miR-150-5p is downregulated during heart failure and acts as an inhibitor of the proapoptotic protein small proline-rich protein 1a, thereby reducing cardiac apoptosis and fibrosis [49].

In previous published studies, the age and history of CV diseases or the presence of CV risk factors were associated with ibrutinib-related CVAEs [6,26,27]. In our study, none of these parameters came out statistically significant. The absence of statistically significant parameters such as age, history of hypertension, and left atrial dilatation or strain is probably explained by the small number of patients included in our cohort. In a prospective cohort of 53 patients treated with ibrutinib, with an AF incidence of 38% over two years of follow-up, left atrial enlargement, defined as a left atrial area > 40 mL/m^2^, was associated with a markedly increased risk of AF with a HR = 8.68 (% CI: 2.86–26.26) [6]. In a retrospective study of 66 patients treated with ibrutinib, 22 developed AF, and several parameters exhibited a significant association with ibrutinib-related AF: E/Ea was significantly higher among the AF group (11.5 vs. 9.3, *p* = 0.04) and peak atrial longitudinal strain was significantly lower among the AF group (30.3% vs. 36.3%, *p* = 0.01) [26].

## 5. Study Limitations

The main limitation of this study is the small size of the cohort, which implies low statistical power and not enough events to undertake multivariate analysis. Indeed, our results are not adjusted on potential confounding factors. Furthermore, biological parameters were assessed only at baseline prior to ibrutinib initiation, as follow-up samples were not available. Additionally, the absence of cardiac tissue precluded direct measurement of CSK activity, limiting our ability to strengthen the evidence supporting the involvement of the CSK–Src–aldosterone–mineralocorticoid receptor axis in the pathophysiology of ibrutinib-related CVAEs. Moreover, baseline use of angiotensin-receptor blockers (ARBs) and ACE inhibitors can clearly influence plasma concentrations of renin and aldosterone. Therefore, these results should be interpreted with caution. This study has great potential to be prospective but this is a monocentric study, which can bias the external validity. Indeed, every patient included in this study is followed in our cardio-oncology program, which allows for specialized management of the specificities of these patients suffering from hematologic malignancies and at risk of CVAEs.

## 6. Conclusions

In this small prospective and monocentric cohort of 25 patients exposed to ibrutinib for B-cell malignancy, 7 patients experienced 9 CVAEs during a median follow-up of 672 days. The CSK–Src–aldosterone–mineralocorticoid receptor axis may represent a potential pathway involved in the development of ibrutinib-related CVAEs, although this remains to be further explored. Furthermore, history of HTN and miR-150-5p levels at baseline were univariably associated with CVAEs occurrence. These results need to be confirmed in larger multicenter cohort studies.

## Figures and Tables

**Figure 1 biomedicines-13-02184-f001:**
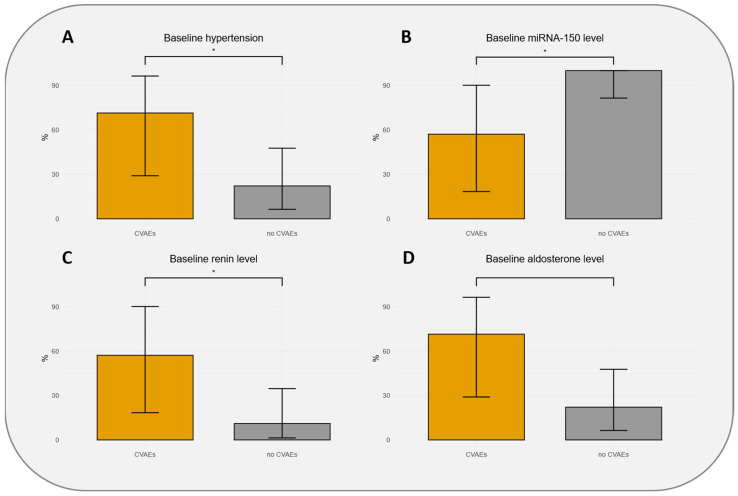
Distributions of population with 95% confidence intervals according to hypertension baseline status (**A**), thresholds of baseline miR-150-5p level (**B**), baseline renin level (**C**), and baseline aldosterone level (**D**). CVAEs: cardiovascular adverse events. * indicates significant association (*p*-value < 0.05).

**Figure 2 biomedicines-13-02184-f002:**
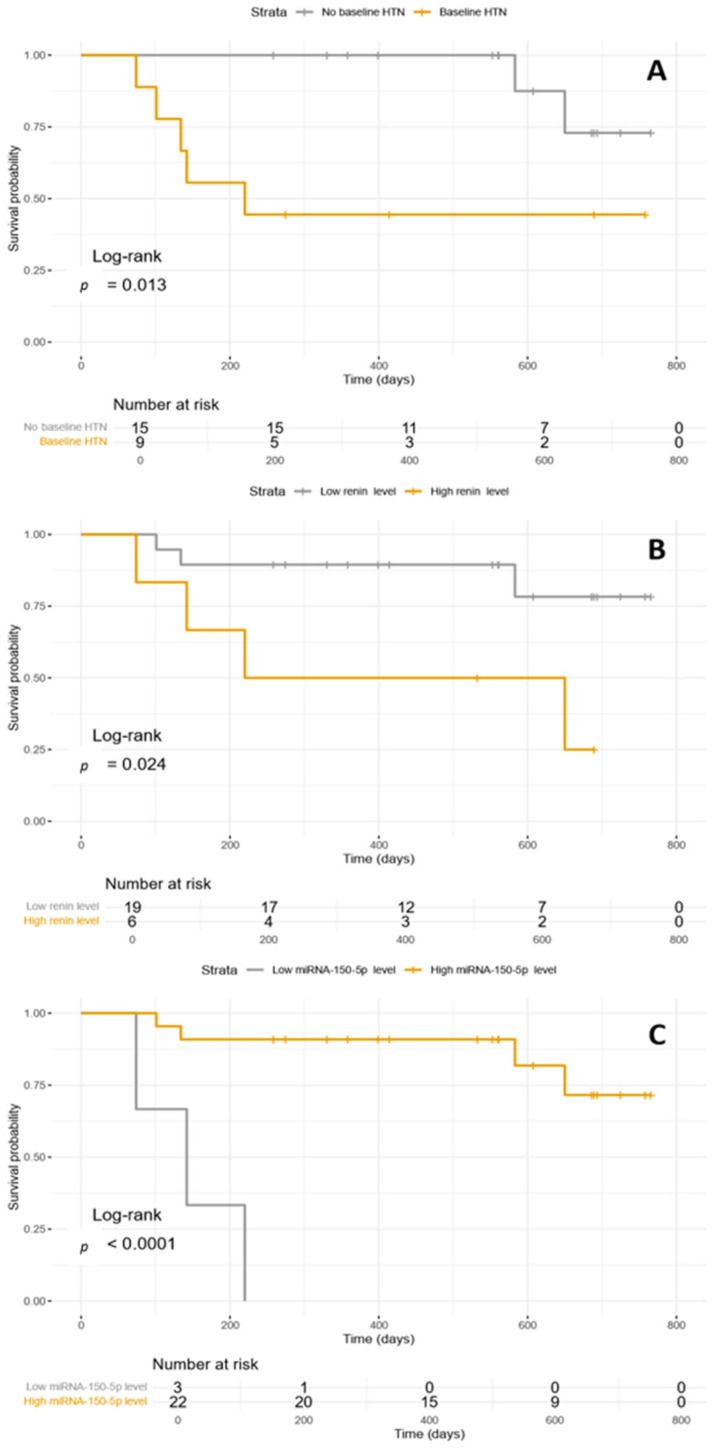
(**A**) Kaplan–Meyer curve illustrating CVAEs occurrence according to baseline HTN; (**B**) Kaplan–Meyer curves illustrating CVAEs occurrence according to baseline plasmatic renin level; (**C**) Kaplan–Meyer curves illustrating CVAEs occurrence according to baseline miR-150-5p level. HTN: hypertension.

**Figure 3 biomedicines-13-02184-f003:**
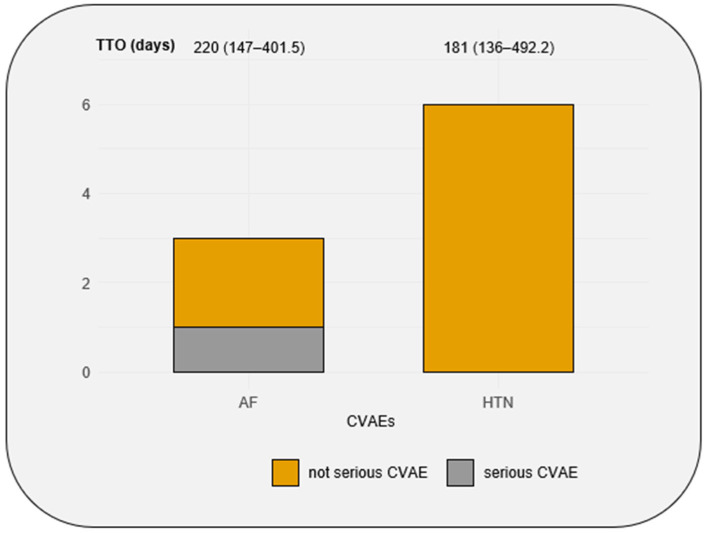
Number of AF and HTN events in the cohort, the seriousness of AF and HTN events, and the median of time to onset of AF and HTN events. CVAEs: cardiovascular adverse events; TTO: time to onset; AF: atrial fibrillation; HTN: hypertension. Time to onset is expressed with median (quartile 1–quartile 3).

**Figure 4 biomedicines-13-02184-f004:**
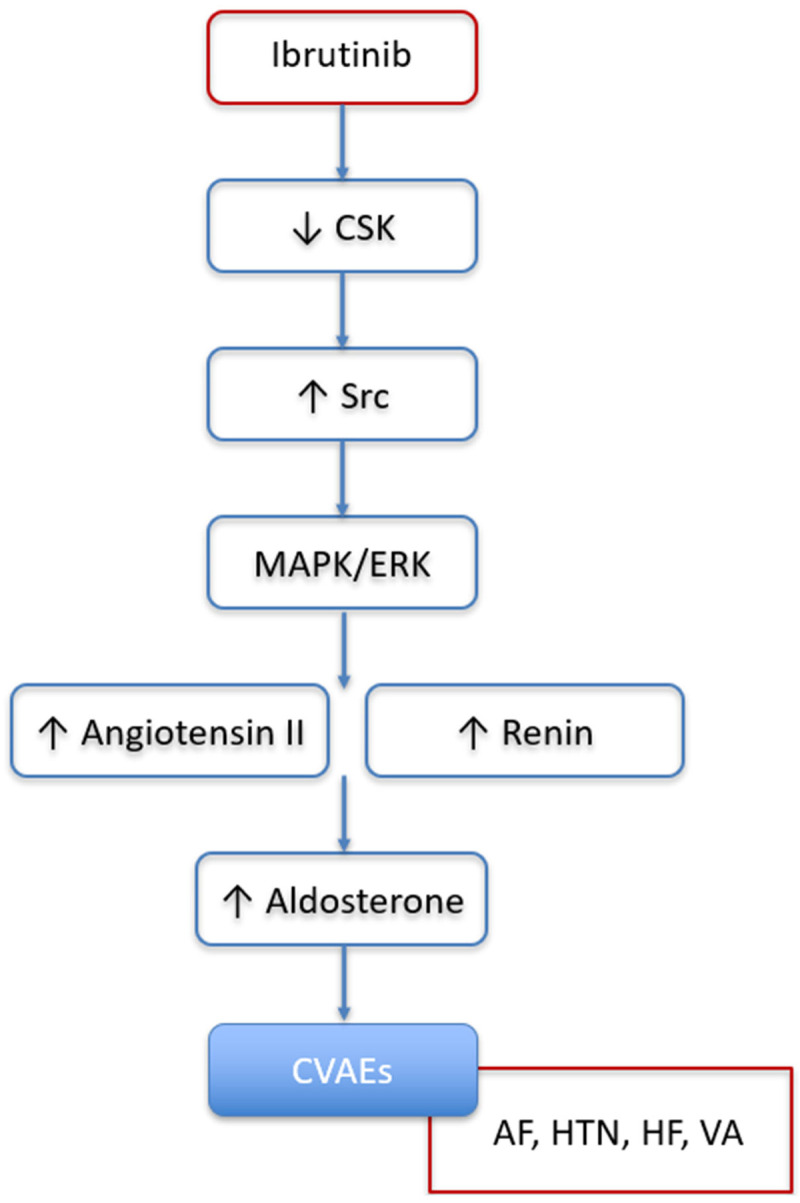
Potential role of renin–angiotensin–aldosterone system, activated by CSK inhibition and subsequent Src activation, in promoting development of ibrutinib-related cardiovascular adverse events. CSK: C-Terminal Src Kinase; CVAEs: cardiovascular adverse events; AF: atrial fibrillation; HTN: hypertension; HF: heart failure; VA: ventricular arrythmias.

**Table 1 biomedicines-13-02184-t001:** Baseline characteristics of cohort. CVAE: cardiovascular adverse event; BMI: body mass index; HTN: hypertension; AF: atrial fibrillation; LBBB: left bundle branch block; RBBB: right bundle branch block; AV: atrio-ventricular; CCB: calcium-channel blocker, LVEF: left ventricular ejection fraction; LA: left atrium; CRP: C reactive protein; TNF: tumor necrosis factor; IL: interleukin; ACE-2: angiotensin-converting enzyme 2.

Characteristics	Total (n = 25)	No CVAE Group (n = 18)	CVAEs Group (n = 7)	*p*-Value
Demographic parameters				
Homme	16 (64%)	10 (55.6%)	6 (85.7%)	0.355
Age	72 (63–77)	73 (63.2–76.8)	69 (65–77)	1
BMI	23.7 (21.9–27.7)	22.9 (20.9–26.6)	24.6 (23.3–29.6)	0.204
HTN	9 (36%)	4 (22.2%)	5 (71.4%)	0.061
Hypercholesterolemia	3 (12%)	2 (11.1%)	1 (14.3%)	1
Diabetes	1 (4%)	1 (5.6%)	0 (0%)	1
Renal insufficiency	0 (0%)	0 (0%)	0 (0%)	
History of AF	1 (4%)	0 (0%)	1 (14.3%)	0.28
History of heart failure	0 (0%)	0 (0%)	0 (0%)	
History of stroke	2 (8%)	1 (5.6%)	1 (14.3%)	0.49
History of peripheral vascular disease	1 (4%)	1 (5.6%)	0 (0%)	1
Cardiac artery disease	3 (12%)	3 (16.7%)	0 (0%)	0.539
Veinous thromboembolism	1 (4%)	1 (5.6%)	0 (0%)	0.49
LBBB, RBBB, 1st degree AV block	6 (24%)	4 (22.2%)	2 (28.6%)	1
Type of hemopathy				
-Chronic lymphoid leukemia	17 (68%)	13 (72.2%)	4 (57.1%)	0.64
-Lymphoma	3 (12%)	3 (16.7%)	0 (0%)	0.534
-Waldenstrom disease	4 (16%)	2 (11.1%)	2 (28.6%)	0.548
Drugs at baseline				
-Aspirin	6 (24%)	4 (22.2%)	2 (28.6%)	1
-Anticoagulant	2 (8%)	1 (5.6%)	1 (14.3%)	0.49
-Statine	5 (20%)	4 (22.2%)	1 (14.3%)	1
-Beta-blocker	4 (16%)	3 (16.7%)	1 (14.3%)	1
-CCB dihydropyridinique	8 (32%)	3 (16.7%)	5 (71.4%)	0.017
-CCB non-dihydropyridinique	1 (4%)	1 (5.6%)	0 (0%)	1
-ACE inhibitors	3 (12%)	2 (11.1%)	1 (14.3%)	1
-Angiotensin-receptor blockers	4 (16%)	3 (16.7%)	1 (14.3%)	1
-Antiarrhythmic drugs	1 (4%)	0 (0%)	1 (14.3%)	0.28
Echographic parameters				
LVEF (%)	64.7 (60.9–68.6)	64 (61.6–67.8)	67.4 (61.3–71)	0.274
LA dilatation	10 (40%)	6 (33%)	4 (57%)	0.314
E wave (cm/s)	74.6 (63.4–82.9)	72.6 (62.8–85.2)	76.7 (70.1–79.2)	0.85
E/A ratio	0.9 (0.7–1.2)	0.9 (0.8–1.1)	0.9 (0.8–1.1)	0.85
E/Ea ratio	9 (7.4–9.8)	8.1 (7.1–9.4)	9.9 (8.3–13.3)	0.083
LA reservoir strain	31.8 (26.7–37.1)	31.1 (26.7–37.1)	33.9 (29.8–40.4)	0.574
LA conduit strain	17 (14.6–23.5)	16.9 (13.9–22)	21.2 (16.4–27.2)	0.395
LA contractile strain	14.3 (10.5–17.2)	14.3 (10.7–17.2)	12.7 (10.2–16.5)	0.698
LA compliance	4.6 (3.1–5)	4.6 (3.7–5)	3.2 (3–3.8)	0.571
Biologic parameters				
Creatinine (µmol/L)	4 (16%)	1 (6%)	3 (43%)	0.053
CRP (ng/mL)	11 (44%)	9 (50%)	2 (29%)	0.407
Galectin-3 (ng/mL)	11 (44%)	9 (50%)	2 (29%)	0.407
Myeloperoxydase (ng/mL)	4 (16%)	4 (22%)	0 (0%)	0.294
Renin (pg/mL)	6 (24%)	2 (11%)	4 (57%)	0.032
Aldosterone (pg/mL)	9 (36%)	4 (22%)	5 (71%)	0.058
TNF-alpha (pg/mL)	3 (12%)	1 (6%)	2 (29%)	0.18
IL-6 (pg/mL)	8 (32%)	4 (22%)	4 (57%)	0.156
ACE-2 (ng/mL)	2 (8%)	1 (6%)	1 (14%)	0.49
Troponin (pg/mL)	4 (16%)	1 (6%)	3 (43%)	0.053
miR-9 (zmol/µL)	3 (12%)	3 (17%)	0 (0%)	0.534
miR-199 (zmol/µL)	6 (24%)	4 (22%)	2 (29%)	1
miR-22 (zmol/µL)	18 (72%)	15 (83%)	3 (43%)	0.066
miR-99 (zmol/µL)	6 (24%)	5 (28%)	1 (14%)	0.637
miR-150-5p (zmol/µL)	22 (88%)	18 (100%)	4 (57%)	0.015
miR-328 (zmol/µL)	19 (76%)	15 (83%)	4 (57%)	0.298

## Data Availability

Data are available for academic purposes upon reasonable request.

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
