# Peer review of "Association Between the Renin–Angiotensin System and Ibrutinib-Related Cardiovascular Adverse Events: A Translational Cohort Study"

_biomedicines, 2025, doi:10.3390/biomedicines13092184_

Round 1

Reviewer 1 Report

Comments and Suggestions for Authors

The authors hypothesize that baseline parameters related to the renin-angiotensin-aldosterone system (RAAS) may be predictive of cardiovascular adverse events (CVAEs) in patients treated with ibrutinib for B-cell malignancies. Specifically, the study aims to evaluate if elevated baseline plasma levels of renin and aldosterone are associated with a higher incidence of CVAEs such as atrial fibrillation, hypertension, heart failure, and ventricular arrhythmias.

The strength of the study is rigorous baseline phenotyping and systematic follow-up.
However the limitation is  small sample size (n=25), limiting the statistical power and precluding multivariate modelling.

Due to sample size, the findings should be interpreted as hypothesis-generating, not conclusive.

The manuscript is well-structured, generally clear, and written in a professional tone.Some grammar issues and formatting inconsistencies (especially in affiliations and table layout) could be improved during copyediting.

Please mention that the study is not powered for multivariate modelling.

Emphasize in the conclusion that findings need replication in larger multicenter cohorts.

Even in univariate analysis, briefly reflect on potential collinearity (e.g., between hypertension and renin).

Please mention the association between inflammatory markers and CVAEs. Please mention the predictive value of albumin in CVAE and the association between albumin and inflammation citing 'Overlap Between Nutritional Indices in Patients with Acute Coronary Syndrome: A Focus on Albumin'.

Reviewer 2 Report

Comments and Suggestions for Authors

The manuscript presents a study of 25 patients on ibrutinib, of whom 7 experienced cardiovascular adverse events during nearly two years of follow-up. The authors found that baseline renin levels and baseline miR-150-5p levels were lower in patients with CVAE. There were no significant baseline differences in left atrial structure and function by echocardiography.

Major Comments

1. They hypothesize that the mechanism for the association with higher renin levels with CVAEs is that ibrutinib inhibits CSK, which, in turn, inhibits an activator of the RAAS — src from the SFK kinase family. This hypothesis is only speculative based on the data they obtained because they did not measure CSK and src in these patients, and they have only one assessment of renin around that time ibrutinib was started. The paper would support this hypothesis better if they provided CSK, src, renin, and aldosterone levels before ibrutinib was started (or around the time it was started) and several months after it was started. Is this possible? If this is not possible, the lack of these measurements should be highlighted in the limitations, and the Discussion should note the speculative nature of the hypothesis that is not fully supported by experimental data.

2. In the description of the proposed CSK-src-RAAS mechanism, the mechanism should be described more accurately. Src activates renin via cAMP, and CSK inhibits src. The related text in the Discussion section should be corrected/improved.

3. In the baseline Table and bar graphs, continuous variables of interest are described as the number with a serum level above a certain threshold. Was this the median value? This needs to be clarified? It is unusual to present continuous variables by groups in this way. I recommend presenting these data as continuous variables with a p-value based on a t-test. It would also be better to present these data in same way in Figure 1 and Figure 2.

4. Was the exploration of miR-150-5p levels purely speculative? Was this based on a pre-study hypothesis? This should be clarified. If they were just exploring possible associations, the statistical analysis and p-values should reflect the multiple comparisons (using a Bonferroni correction or a less conservative test).

5. Some patients were on ACE inhibitors or ARBs at baseline. Did patients continue these medications during follow-up. Did some patients start taking these medications during follow-up? As these medications interact directly with the RAAS, there should be some consideration about them in the analysis and discussion.

6. Presumably, the time to the first CVAE is available. In this case, the paper should show Kaplan-Meier curves stratified based on the median renin value at baseline for time to first CVAE with the log-rank test p-value. A hazard ratio and associated p-value from a Cox proportional hazards model would also be of interest. The same procedure should be followed using the median value of the miR-150-5p level.

7. The manuscript text states that miR-150-5p levels were lower in patients who did not have CVAEs, which is incorrect. Both Table 1 and Figure 1 show that miR-150-5p levels were lower in patients who did have CVAEs. This also makes the explanation provided in the Discussion for this association invalid. This inconsistency needs to be addressed.

Minor comments:

1. The font size in Table 1 should be reduced, as the columns run together.

2. The citations should have a superscript.

3. Figure 3 should be updated based on the comments above. 

4. The paper uses “miR-150” in Table 1 and the figures but “miR-150-5p” in the text. Usage should be consistent.
